# Sequence-sensitive elastic network captures dynamical features necessary for miR-125a maturation

**Olivier Mailhot**[1,2,3,4], **Vincent Frappier**[5], **François Major**[2,3]*, **Rafael J. Najmanovich**[4]*

**1** Department of Biochemistry and Molecular Medicine, Université de Montréal, Montreal, QC, Canada, **2** Department of Computer Science and Operations Research, Université de Montréal, Montreal, QC, Canada, **3** Institute for Research in Immunology and Cancer, Université de Montréal, Montreal, QC, Canada, **4** Department of Pharmacology and Physiology, Université de Montréal, Montreal, QC, Canada, **5** Generate Biomedicines, Cambridge, Massachusetts, United States of America

* francois.major@umontreal.ca (FM); rafael.najmanovich@umontreal.ca (RJN)

**Data Availability Statement:** All relevant data are within the manuscript and its Supporting information files.

## Abstract

The Elastic Network Contact Model (ENCoM) is a coarse-grained normal mode analysis (NMA) model unique in its all-atom sensitivity to the sequence of the studied macromolecule and thus to the effect of mutations. We adapted ENCoM to simulate the dynamics of ribonucleic acid (RNA) molecules, benchmarked its performance against other popular NMA models and used it to study the 3D structural dynamics of human microRNA miR-125a, leveraging high-throughput experimental maturation efficiency data of over 26 000 sequence variants. We also introduce a novel way of using dynamical information from NMA to train multivariate linear regression models, with the purpose of highlighting the most salient contributions of dynamics to function. ENCoM has a similar performance profile on RNA than on proteins when compared to the Anisotropic Network Model (ANM), the most widely used coarse-grained NMA model; it has the advantage on predicting large-scale motions while ANM performs better on B-factors prediction. A stringent benchmark from the miR-125a maturation dataset, in which the training set contains no sequence information in common with the testing set, reveals that ENCoM is the only tested model able to capture signal beyond the sequence. This ability translates to better predictive power on a second benchmark in which sequence features are shared between the train and test sets. When training the linear regression model using all available data, the dynamical features identified as necessary for miR-125a maturation point to known patterns but also offer new insights into the biogenesis of microRNAs. Our novel approach combining NMA with multivariate linear regression is generalizable to any macromolecule for which relatively high-throughput mutational data is available.

## Author summary

Ribonucleic acids (RNAs) are biomolecules which play essential roles in the function of all living organisms. These molecules can adopt defined 3D structures in the cell, but they

**Funding:** This work was supported by Natural Sciences and Engineering Research Council of Canada (NSERC) Discovery program grants (FM and RJN); Genome Canada and Genome Quebec (RJN); Compute Canada (RJN) and Canadian Institutes of Health Research (CIHR) (FM, grant number MOP-93679). The funders played no role in the study design, data collection and analysis, decision to publish, or preparation of the manuscript.

**Competing interests:** The authors have declared that no competing interests exist.

also move around their equilibrium structure. RNA function is intimately related to structural dynamics, which, however, can be costly to simulate. In the present study, we adapt a fast method for the computational study of protein dynamics, called ENCoM, to work on RNA molecules. We benchmark its performance against other similar methods and find that ENCoM has a clear advantage when it comes to predicting large-scale dynamics. Moreover, ENCoM is unique in its ability to predict the effect of mutations on structural dynamics, as was already shown for proteins. This ability extends to RNA: we capture dynamics-function relationships apparent from experimental maturation efficiency data on over 26 000 sequence variants of a human microRNA, miR-125a. These dynamics-function relationships are learned by a novel linear model combining the reduced ENCoM dynamical information and the energy of folding. The low computational cost of this technique opens up the possibility of high-throughput prediction of RNA and protein functional properties from sequence information, if starting structures are known or can be predicted.

## Introduction

It is now well-established that ribonucleic acid (RNA) molecules possess a diverse range of functions in all domains of life and there is growing interest in understanding and characterizing these functions [1]. However, our structural knowledge of RNA is still scarcer than that of proteins, the latter being based on about two orders of magnitude more structures in the Protein Data Bank (PDB) [2]. RNAs are dynamic and thus exist in an ensemble of conformations that are intimately tied to their function [3]. As is the case with proteins, point mutations can disrupt the function of RNAs by affecting their structural dynamics while leaving their equilibrium structure intact [4]. In order to study dynamical effects from immense numbers of such theoretical mutations, fast computational prediction of RNA conformational dynamics from known or predicted three-dimensional (3D) structures is a necessity. In the case of proteins, a popular method for obtaining such information is coarse-grained normal mode analysis (NMA) using elastic network models (ENMs) [5]. Many different ENMs with various application niches have been developed for proteins, while the few studies scpecifically investigating RNA [6–8] have used either the Gaussian Network Model (GNM), which predicts isotropic fluctuations [9, 10], or the Anisotropic Network Model (ANM) which predicts directional motions [11].

In the last years, we introduced the Elastic Network Contact Model (ENCoM) for NMA of proteins [12–14]. ENCoM was shown to perform significantly better than other ENMs for the prediction of conformational changes. More importantly, to our knowledge ENCoM is the only coarse-grained NMA model aware of the all-atom context of residues, making it sensitive to the effect of mutations even when they do not change the backbone geometry of the protein [13, 15]. In the present work, we adapted ENCoM to work on RNA molecules and benchmarked its performance on a set of 480 curated RNA structures derived from the RNA-only structures available in the PDB resolved by either nuclear magnetic resonance (NMR) or X-ray crystallography. ENCoM outperforms ANM on the prediction of conformational changes derived from pairs of X-ray structures and conformational variance within NMR ensembles but performs slightly worse on experimental B-factors prediction, as previously observed in the case of proteins [12].

NMA, whether coarse-grained or all-atom, is an analytical technique that can be used to explore the entire conformational space of a macromolecule at every timescale, often with an

associated computational cost tremendously lower than that of, for example, molecular dynamics (MD) simulations [16]. In NMA, a molecule is represented as a system of beads connected by springs with a harmonic potential. For a system of N beads, the normal modes are the 3N eigenvectors of the Hessian matrix, each with its associated eigenvalue. The first six normal modes are trivial motions which represent the system's translational and rotational degrees of freedom. From the seventh, the modes are ordered with increasing frequency and thus increasing energetic cost to achieve the same amount of deformation. Usually, a small set of the first non-trivial modes is sufficient to describe global motions of biological significance, for instance in the case of the hinge-bending motion of the citrate synthase enzyme [17].

In the study of proteins, coarse-grained ENMs generally use one bead per amino acid, located at the $C_\alpha$ atom [5]. Efforts have been made to expand these coarse-grained models to include more information, such as the $C_\beta$ atom [18], or the combination of different levels of coarse-graining [19, 20]. However, single-bead per residue models continue to be the standard for coarse-grained NMA of proteins due to their good preservation of the slowest and most global motions. The two most widely-used models for proteins are arguably ANM [21] and GNM [6] and both use one bead per residue, situated at the $C_\alpha$ atom by default. By contrast, when these models were applied to RNA, they performed significantly better when three beads were assigned for each nucleotide: one for the phosphate group, one for the sugar group and one for the nucleobase (Fig 1A and 1B) [6–8]. This improvement can be explained in part by the higher intrinsic flexibility of RNA, having six backbone torsional angles per residue instead of the three found in proteins [22], and by the importance of base pairing in forming specific 3D architectures.

Mature microRNAs (miRs) are small non-coding single-stranded RNAs of about 22 nucleotides. They regulate gene expression by guiding RNA-Induced Silencing Complexes (RISCs) to complementary regions within messenger RNA (mRNA) sequences, triggering silencing of these targets [23]. The canonical pathway of miR production starts with their primary transcript, which forms a hairpin-loop structure of approximately 35 base-pairs (pri-miR). A crucial step is the recognition of the pri-miR by the DROSHA/DGCR8 heterotrimer, also called Microprocessor, which cleaves the hairpin around the 11th base-pair from the bottom [24].

A single nucleotide polymorphism (SNP) in human miR-125a predisposes to aggressive types of breast cancer. One of the effects of the SNP (G22U) is to prevent cleavage by the Microprocessor [25]. The SNP does not affect the global structure of the minimum free energy (MFE) state. Moreover, we showed that changes in 2D structural dynamics of the 16 possible base pairs at the SNP position correlate well with their maturation efficiency measured from cellular luciferase assays [4].

David Bartel's group has generated high-throughput maturation efficiency data for upwards of 50 000 pri-miR-125a sequence variants using an enzymatic assay with purified Microprocessor [26]. We generated all pri-miR-125a variants *in silico* and studied their 3D structural dynamics with both ENCoM and ANM to test whether ENCoM's ability to predict the effect of mutations on function extends to RNA. We found that ENCoM's sequence sensitivity allows it to capture variance from the miR-125a maturation dataset even when no sequence features are shared between training and testing sets, a capacity which confirms sequence-dynamics-function relationships exist as part of the signal recognized by the Microprocessor for microRNA production. As expected, ENCoM is the only tested model exhibiting this ability, evidence that we have successfully extended to RNA ENCoM's unique sensitivity to the effects of mutations [12, 13, 15].

We also introduce as part of the present work what is, to the best of our knowledge, the first high-throughput use of dynamical information inside machine learning models (regularized linear regression). The low computational cost of coarse-grained NMA opens the door to

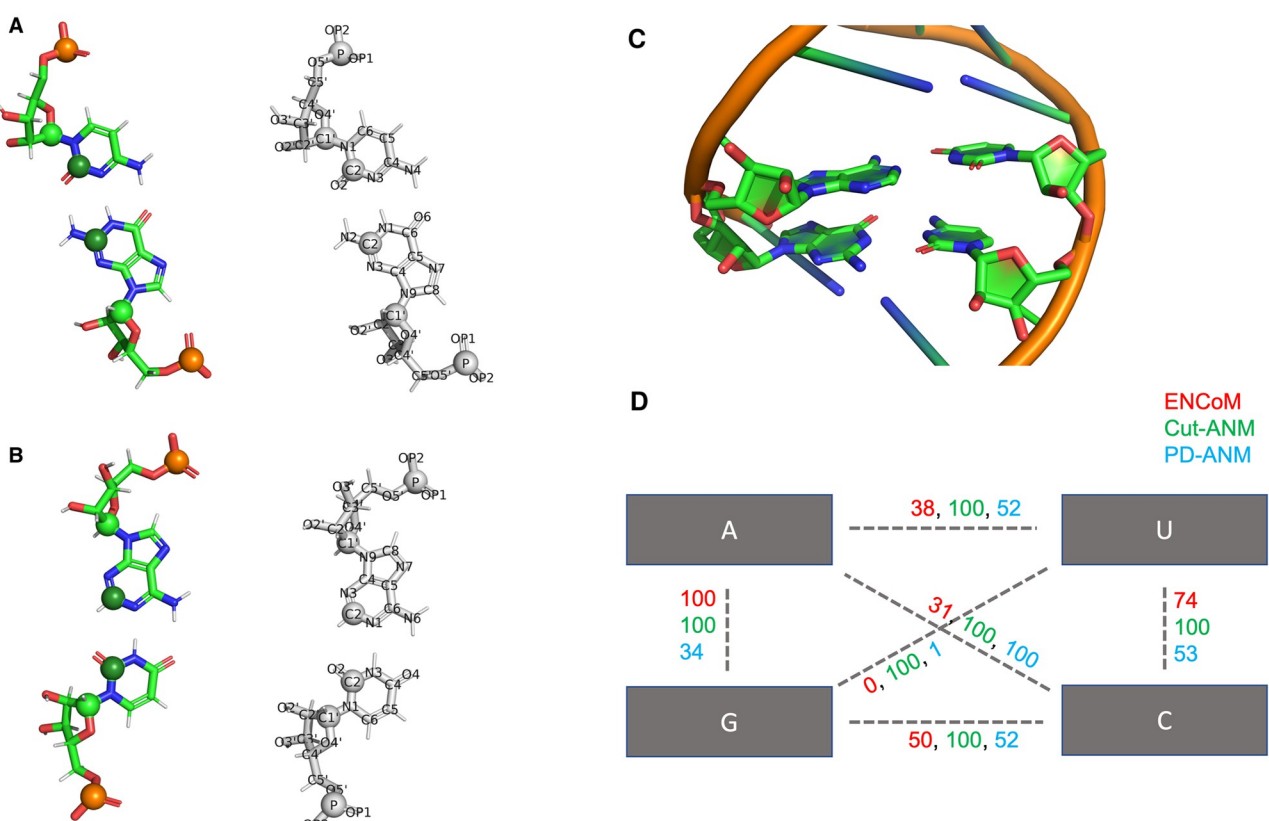

**Fig 1. Assignment of beads on the four standard nucleotides and strength of base-base interactions in the different models.** A-B) Phosphate atoms are in gold, C1' carbons from the sugar group in light green and C2 carbons from the nucleobase in dark green. A GC pair is shown in A) and an AU pair in B). The atom names are shown over white structures to enhance legibility. The assignation of the ENCoM atom types to each atom name is given in Table H in S1 Text. All three beads are included in the ENMs tested for the present work. The base pairs shown were extracted from an A-form RNA helix generated using the MC-Fold and MC-Sym pipeline [36]. C) A stack of canonical base pairs, AU over GC, extracted from PDB code 2V6W. ENCoM's $\beta_{ij}$ term, Cut-ANM's binary interaction and PD-ANM's distance dependent interaction are shown in panel D) between all pairs of nucleobase beads, from the exact structure shown in C). The parameters used are 10 Å cutoff for Cut-ANM and a power dependency of 7 for PD-ANM, the optimal values found in the present study for the B-factors benchmark. The values are rescaled to a maximum of 100 to allow better comparison of the three potentials. The cytosine is in close proximity to the diagonal adenosine, hence the high interaction value assigned by PD-ANM, proportional to the inverse of the distance to the $7^{\text{th}}$ power.

ultra-high-throughput prediction of variant effects. A recent review of machine learning-based variant effect predictors, published in April 2022 by Horne and Shukla, noted the need for such tools able to scale structural dynamics predictions to millions of theoretical variants [27]. Our work can be seen as a proof of concept that ENCoM can fill this currently void niche.

Finally, when mapping the coefficients of the linear model back to the miR-125a structure, the most salient feature is backbone flexibility at the UC noncanonical ninth base pair. This pattern corresponds to a motif known to enhance microRNA processing, the mismatched GHG [26]. The identification of a known biological feature from the computed flexibility at each position points to the strength of the approach we present for driving new hypotheses about dynamics-function relationships in biomolecules.

## Materials and methods

### Elastic network models

The perhaps counterintuitive notion that global biomolecular motions can be captured almost as well by very simplified potentials as by all-atom normal mode analysis was championed in

Monique Tirion's seminal work, which can be considered the birth of coarse-grained normal mode analysis [28]. Early studies have hypothesized this surprisingly high qualitative performance of coarse-grained NMA is due to the fact that global motions are mostly dictated by the geometry of the studied biomolecule [29]. Based on the geometry hypothesis, Lin and Song developed the generalized Spring Tensor Model (STeM), maintaining the popular $C_\alpha$ coarse-graining but replacing the distance-based interaction scheme of ANM by a four-term Gō-like potential: covalent bond stretching, angle bending, dihedral angle torsion and non-bonded (or long-range) interactions [30]. The ENCoM potential is based on the STeM potential, of which the non-bonded interaction term is then modulated according to all-atom interactions between beads that are close in space [12]. The terms of the potential are at the level of entire residues: a "covalent" bond describes the interaction between sequential amino acids in proteins. In the present case of a three-bead RNA model, "covalent" interactions happen between phosphate-sugar and sugar-base bead pairs, as is naturally derived from the all-atom connectivity of RNA. Similarly, "angles" describe bending happening at the level of groups of three sequentially connected beads, and "dihedrals" in groups of four. As a harmonic potential, the ENCoM potential describes the energy of an arbitrary conformation $\vec{R}$ around the input conformation, $\vec{R_0}$, which by default has zero energy in the NMA representation:

$$
V_{\text{ENCoM}}(\vec{R}, \vec{R_0}) = \sum_{\text{bonds}} V_1(r, r_0) + \sum_{\text{angles}} V_2(\theta, \theta_0)
$$

$$
+ \sum_{\text{dihedrals}} V_3(\phi, \phi_0) + \sum_{i<j-3} V_4(r_{ij}, r_{ij_0})
$$

$$
= \sum_{\text{bonds}} \alpha_1(r - r_0)^2 + \sum_{\text{angles}} \alpha_2(\theta - \theta_0)^2 \qquad (1)
$$

$$
+ \sum_{\text{dihedrals}} \left[ \alpha_3(1 - \cos(\phi - \phi_0)) + \frac{\alpha_3}{2}(1 - \cos 3(\phi - \phi_0)) \right]
$$

$$
+ \sum_{i<j-3} (\beta_{ij} + \alpha_4) \left[ 5\left(\frac{r_{ij_0}}{r_{ij}}\right)^{12} - 6\left(\frac{r_{ij_0}}{r_{ij}}\right)^{10} \right]
$$

In the ENCoM potential, the non-bonded interaction term is modulated according to the surface area in contact between residues by the $\beta_{ij}$ term:

$$
\beta_{ij} = \sum_{k}^{N_i} \sum_{l}^{N_j} \epsilon_{T(k)T(l)} S_{kl} \qquad (2)
$$

$\epsilon_{T(k)T(l)}$ represents the interaction between atoms of types $T(k)$ and $T(l)$ while $S_{kl}$ is the surface area in contact between the two atoms, calculated using a constrained Voronoi procedure, as described by McConkey and co-workers [31]. This Voronoi procedure is very fast, yet takes all proximal atoms into account when evaluating the surface area in contact between a given pair of atoms. Moreover, the van der Waals radii are extended by the approximate radius of a water molecule (1.4 Å) to form what is termed the extended contact radius. This radius extension allows the implicit consideration of solvent-mediated contacts as two atoms can have a surface area in contact up to a distance given by the sum of their extended radii.

The atom types used are those of Sobolev and co-workers [32], which are divided in eight classes: hydrophilic, acceptor, donor, hydrophobic, aromatic, neutral, neutral-donor and neutral-acceptor. Table H in S1 Text shows the assigned type of every atom from the four common ribonucleotides. The interaction between two types, $\epsilon_{T(k)T(l)}$, is binary: either favorable or

unfavorable. The interaction matrix has been described by Sololev and co-workers [32] and was neither altered for the present work nor during the original development of ENCoM [12].

Each of ENCoM's four potential terms is modulated by a weight coefficient; these four coefficients together represent the four parameters of the model and were explored extensively in the past. ENCoM's performance came out as very robust across a wide range of parameters [12]. For the present study, we thus used exactly the same parameters as those published for proteins. This makes ENCoM in its present form readily applicable to proteins, RNA and RNA-protein complexes without the need for re-parameterization. Let us remind the pseudo-physical nature of all coarse-grained normal mode analysis models. The parameters described here are much more related to the relative importance of each term than to any measurable quantity, as opposed to molecular dynamics force field parameters. Thus, while parameter search is a worthwhile endeavor, we leave it for future work and are satisfied for now with a unique parameter set that works across both protein and RNA molecules. The four weight parameters are the following: $\alpha_1 = 10^3$, $\alpha_2 = 10^4$, $\alpha_3 = 10^4$, $\alpha_4 = 10^{-2}$. Favorable interactions have a weight of 3 and unfavorable interactions a weight of 1 [12].

The Anisotropic Network Model (ANM) uses a simple potential where all pairs of beads closer than a given distance cutoff are considered to be interacting and are connected with a spring of uniform constant $\gamma$ [33]. The interaction distance cutoff $R_c$ (generally set to 18 Angstroms (Å) for proteins) is thus the single parameter of the model, which we have extensively tested in increments of 0.5 Å for the B-factors prediction benchmark. The higher and lower bounds for the cutoffs tested were chosen to show a clear decreasing trend in either direction, confirming that the value for maximal performance has been found.

In addition to the classical cutoff-based ANM, ANMs using spring constants that are distance-dependent have been introduced [34]. We also tested this version of the model, called power dependence ANM (PD-ANM), as it was applied to RNA by Zimmermann and Jernigan [7]. Instead of a distance cutoff parameter, all beads are connected by springs in this model. The spring constant varies according to the inverse distance to the $x^{th}$ power, where $x$ is the power dependence, the only parameter of the model. As we did for the cutoff-based ANM (Cut-ANM), we scanned the power dependency parameter in increments of 0.5 for the B-factors benchmark, again making sure to identify a clear decreasing trend in either direction.

For proteins, it is generally believed that ENMs using only one bead per residue situated on the $C_\alpha$ atom still capture the essential low-frequency motions of the molecule [5]. However, since RNA molecules are intrinsically more flexible per residue than proteins, using more beads per residue leads to an increase in the predictive power of the model [6]. For all tested elastic networks, we use a three beads per residue coarse-graining scheme, where we assign one bead each for the sugar, base and phosphate groups positioned on the C1', C2 and P atoms, respectively. This coarse-graining approach was already validated in the case of an ANM [8], thus we employ it for all the models tested thereafter in the present work. Fig 1 shows the positioning of the beads in the four standard nucleotides in panels A and B. Panels C and D illustrate the relative interaction strengths of the three models when looking at base-base interactions from two stacked canonical base pairs: an AU pair on top of a GC pair. ENCoM is the only model capturing the higher rigidity of the GC pair compared to the AU pair. It also captures the lower stacking energy (higher stability) of a purine-purine stack as compared to a pyrimidine-pyrimidine stack. A detailed list of $\beta_{ij}$ terms for all base pairs found in miR-125a is given in Table N in S1 Text.

The three elastic network models tested in this study were ran using the NRGTEN Python package [35] in which they are all implemented by extending the base ENM class. This ensures there are no discrepancies in the assignment of beads or parsing of PDB files since all these steps are performed by the exact same code for all models. The NRGTEN code and

documentation can be accessed at https://nrgten.readthedocs.io and now includes an example on miR-125a (https://nrgten.readthedocs.io/en/latest/rna.html).

## Dataset of RNA structures

Experimental determination of RNA structure is achieved by many different techniques, of which two are by far the most common: X-ray crystallography [37] which takes a snapshot of the molecule near its lowest energy conformation and solution nuclear magnetic resonance (NMR) [38, 39] in which an ensemble of probable conformations is resolved. In the PDB as of 2022–01–08, there were 982 structures solved by X-ray crystallography and 417 NMR ensembles when restricting the search to entries containing only RNA. We used these data in our benchmarks after applying appropriate filters (described at the end of the current section) in order to compare ENCoM's performance to two popular versions of ANM, cutoff-based and power dependent, in the prediction of B-factors, conformational transitions and NMR ensemble conformational variance. We kept all chains present in the biological assembly for X-ray structures and all chains in the structural ensemble for NMR experiments.

Our model requires the manual assignment of atom types for every residue we want to consider so that the surface in contact term can be adequately computed. It is thus possible in theory to include every modified residue. In order to simplify the present study, we chose to restrict the model to the four standard ribonucleotides, replacing all modified ribonucleotides with their unmodified analog using the ModeRNA software [40], which contains information about the 170 modifications present in the MODOMICS database [41]. The atom type assignations for these four standard ribonucleotides are given in Table H in S1 Text. Modified nucleobases were only present in the conformational change benchmark, since they were absent from all gathered NMR ensembles and we specifically excluded structures containing modified nucleobases from the B-factors benchmark. The modified nucleobases initially represented 14% of nucleobases across all structures in the conformational change benchmark. Given the coarse-grained nature of our model, we deem this proportion sufficiently low for our purposes.

We also used ModeRNA to add missing atoms where it was the case, for example terminal phosphate groups, so that each residue was complete. This addition ensured that the assignment of the beads to the residues was standard for all residues. The structures for which ModeRNA produced an error or which produced an error for any of the models tested subsequently were removed from the analysis. Moreover, the size of the X-ray resolved RNA structures was restricted to below 300 nucleotides to reduce computational cost. In the case of the NMR resolved RNAs, no size threshold was applied as no molecules bigger than 155 residues have been elucidated this way. Since we were interested in conformational ensembles solved by NMR, we restricted our analysis to submissions containing at least two models. The list of all PDB codes of the structures kept for the three benchmarks is given in Table I in S1 Text (38 structures for the B-factors benchmark, 129 structures for the conformational change benchmark, 313 NMR ensembles for the structural variance benchmark). The size distribution of the structures kept for the various benchmarks is given in Fig D in S1 Text.

## Sequence clustering

We performed complete linkage clustering [42] on the sequences from our database of structures with a distance threshold of 0.1, ensuring that all pairs of sequences within a given cluster are at least 90% similar. The similarity metric used for calculating distance is derived from the score of a Needleman-Wunsch global alignment [43] between the two sequences, with the

following scoring scheme: gap penalty: -1, mismatch penalty: -1, match score: 1.

$$\text{distance}(s, t) = \frac{\text{score}(s, t)}{\min[\text{length}(s), \text{length}(t)]} \tag{3}$$

For all reported benchmark metrics, we first normalize by sequence cluster before reporting the mean metric. This ensures that molecules which have a lot of conformations sampled in the PDB do not drive the score and instead each RNA or family of RNAs has an equal weight.

## B-factors prediction

Experimental B-factors can be predicted from calculated normal modes as mean-square fluctuations (MSF) of individual residues [11]:

$$\text{MSF}_i = \sum_{n=7}^{3N} \frac{E_{n,i,x}^2 + E_{n,i,y}^2 + E_{n,i,z}^2}{\lambda_n} \tag{4}$$

where $E_{n,i}$ represents the *xyz* displacement of bead $i$ in the $n^{th}$ eigenvector and $\lambda_n$ the associated eigenvalue of that eigenvector. We also define the Entropic Signature, which scales the square fluctuations by the vibrational entropy of each normal mode:

$$\text{ES}_i = \sum_{n=7}^{3N} S_{\text{vib}n}(E_{n,i,x}^2 + E_{n,i,y}^2 + E_{n,i,z}^2) \tag{5}$$

$$S_{\text{vib}n} = \frac{\beta v_n}{e^{\beta v_n} - 1} - \ln(1 - e^{-\beta v_n}) \tag{6}$$

$$v_n = \frac{1}{2\pi} \sqrt{\lambda_n} \tag{7}$$

where $\beta$ is a thermodynamic scaling factor which allows for varying the relative contributions of high- and low-frequency normal modes to the fluctuations of the individual beads.

The Pearson correlation coefficient is then calculated between the predicted and experimental B-factors. In addition to a resolution filter of 2.5 Å or better, we used only the structures for which no modified nucleobases were initially present and in which all residues were complete. This restriction was applied to ensure that every atom used in the prediction had a corresponding experimental B-factor, and not an extrapolated value from the rebuilding of the modified nucleobases and missing atoms by ModeRNA. The 38 remaining structures were clustered according to their sequence as described, giving us 34 sequence clusters, and we computed the mean correlation normalized by cluster.

## Conformational change prediction

The overlap between two conformations is a measure of the similarity between an eigenvector $\vec{E}_n$ predicted using the start conformation and the displacement vector $\vec{R}$ calculated between the coordinates of the target and start conformations, after both have been superimposed [17, 44].

$$\text{overlap}(\vec{E}_n, \vec{R}) = \frac{|\vec{E}_n \cdot \vec{R}|}{\|\vec{E}_n\|\|\vec{R}\|} \tag{8}$$

For each eigenvector, it has a value between 0 and 1, which describes how well we can reproduce the target conformation by deforming the start conformation along the eigenvector. The

cumulative overlap (CO) is also a value between 0 and 1 and describes how well a set of orthogonal motions (eigenvectors) can collectively reproduce the target conformation from the start conformation.

$$CO = \sqrt{\sum_n \text{overlap}(\vec{E}_n, \vec{R})^2} \tag{9}$$

To ensure that the same conformation is not sampled twice, we reject pairs of conformations for which the root mean squared deviation (RMSD) of the center atoms of each bead is lower than 2 Å. Clustering was performed as described, leaving us with 129 unique structures forming 240 pairs of conformations with RMSD greater than or equal to 2 Å, divided in 29 sequence clusters. While an RMSD of 2 might seem low, many biologically relevant motions happen at relatively low RMSDs. Moreover, selecting a lower cutoff allowed us to include more RNA families in the analysis, which is crucial given the scarcity of experimentally resolved RNA structures. High RMSD values (up to 26 Å) were still part of the analysis, as can be seen in Table K in S1 Text. We calculated the cumulative overlap for up to 10% of normal modes both ways for every pair and report the mean cumulative overlap normalized by cluster.

## Non-rotational-translational principal component analysis

Principal component analysis (PCA) is a statistical technique used to transform observations of variables that may be correlated into linearly uncorrelated variables which are called principal components (PCs) [45]. PCA was used here to extract dominant motions apparent within the conformational ensembles obtained from NMR experiments. Each conformation is represented as a vector of length 3N where N is the number of beads in the system (3 per nucleotide). PCA was computed on these vectors using a singular value decomposition (SVD) algorithm [46]. The PCs obtained are analogous to normal modes in that they are the eigenvectors of the covariance matrix of the 3N coordinates from the ensemble of structures. The first PC describes the largest proportion of the variance in the ensemble, and each subsequent component captures the largest proportion of the remaining variance and is orthogonal to the preceding components. However, when there are more than two conformations in the ensemble, rotational and translational motions can be present in the principal components. To our knowledge, this artifact has not been discussed in the context of PCA applied to ensembles of 3D conformations of biomolecules. We thus introduce a correction to obtain principal components without rotational and translational degrees of freedom. First, standard PCs are computed from the ensemble, with all conformations superimposed to the first model. Then, the first six rotational and translational normal modes from that first model are used as the starting basis for Gram-Schmidt orthonormalization of the PCs. This transformation ensures all rotational-translational motions from the PCs are removed and all relevant internal motions are maintained. However, the proportion of variance explained needs to be corrected for each PC according to the amount of rotational-translational variance initially present:

$$c_i = \frac{v_i \left(1 - \sqrt{\sum_{j=1}^6 (\vec{PC}_i \cdot \vec{RT}_j)^2}\right)}{\sum_n c_n} \tag{10}$$

where $v_i$ is the initial proportion of variance explained by $\vec{PC}_i$, $c_i$ is the corrected proportion of variance explained, and $\vec{RT}_j$ are the six rotational-translational normal modes of the first model in the structural ensemble. The PCs are then reordered in decreasing order of corrected

proportion of variance explained. We call this method nrt-PCA (non-rotational-translational PCA) and whenever we refer to PCA or PCs, it is implied that nrt-PCA is used. It ensures the accurate comparison of non-trivial normal modes computed from a representative structure with strictly internal motions apparent from the ensemble. nrt-PCA is has been added to the NRGTEN metrics and documented online [35].

## Structural ensemble variance prediction

The root mean square inner product (RMSIP) is a value between 0 and 1 representing how well a set of normal modes can reproduce the motions from a set of PCs:

$$\text{RMSIP} = \sqrt{\frac{1}{N}\sum_{i=1}^{N}\sum_{j=1}^{M}(\vec{E}_i \cdot \vec{\text{PC}}_j)^2} \tag{11}$$

where $N$ normal modes are compared against $M$ PCs.

The normalized cumulative overlap (NCO) between the first $N$ normal modes and the first $M$ PCs is given by:

$$\text{NCO} = \sum_{j=1}^{M}\left[v_j\sqrt{\sum_{i=1}^{N}\text{overlap}(\vec{E}_i, \vec{\text{PC}}_j)^2}\right] \tag{12}$$

where $\vec{E}_i$ is normal mode $i$, $\vec{\text{PC}}_j$ is principal component component $j$ and $v_j$ is the proportion of variance explained by component $j$. NCO ensures a value between 0 and 1 as both the normal modes and the components are orthogonal, and $v_j$ sums to 1 over all the components.

For the current study, we reported all NCOs and RMSIPs using from 1 normal mode to 10% of the 3N modes, compared against PCs explaining 99% of the observed variance in the ensemble. The normal modes were computed from the conformation of minimal energy in each NMR ensemble, which by convention is the first one in the experimenters' submission. Clustering was performed as described and we computed the mean NCO/RMSIP normalized by cluster.

## Dataset of miR-125a mutations

The Fang *et al.* dataset of high-throughput miR-125a maturation efficiency contains all possible mutations for thirteen 6-nucleotide boxes plus all fifteen possibilities at the 2-nucleotide bulge, which in addition to the major allele sequence (WT) amount to 53 251 sequence variants (Fig 2A). We submitted all sequence variants to 2D structure prediction using the accelerated MC-Flashfold implementation [47] of the MC-Fold software [36]. We found that 29 478 of them adopt the WT 2D minimum free energy (MFE) structure (Fig 2A). Because NMA assumes the input structure is at equilibrium, we decided to restrict the analysis to these 29 478 sequence variants. Furthermore, we realized that the first eight mutated boxes account for the vast majority (over 90%) of these variants which adopt the WT MFE structure. We thus further restricted our analysis to this set of 26 960 variants. The proportion of mutations at each position leading to the WT MFE structure is plotted in Fig 2B, showing a very clear drop in the proportion of sequence variants adopting the WT MFE structure beyond box 8.

Since the mutations were performed exhaustively for each box, there is a great deal of sequence redundancy in the dataset. For example, every of the three possibilities of mutation at a given position appears in as much as 1024 sequence variants in the full dataset. To prevent the learning of sequence features by the models, which could be guessed by the slight variations in the position of the nucleobase beads, we constructed two benchmarks: a hard benchmark

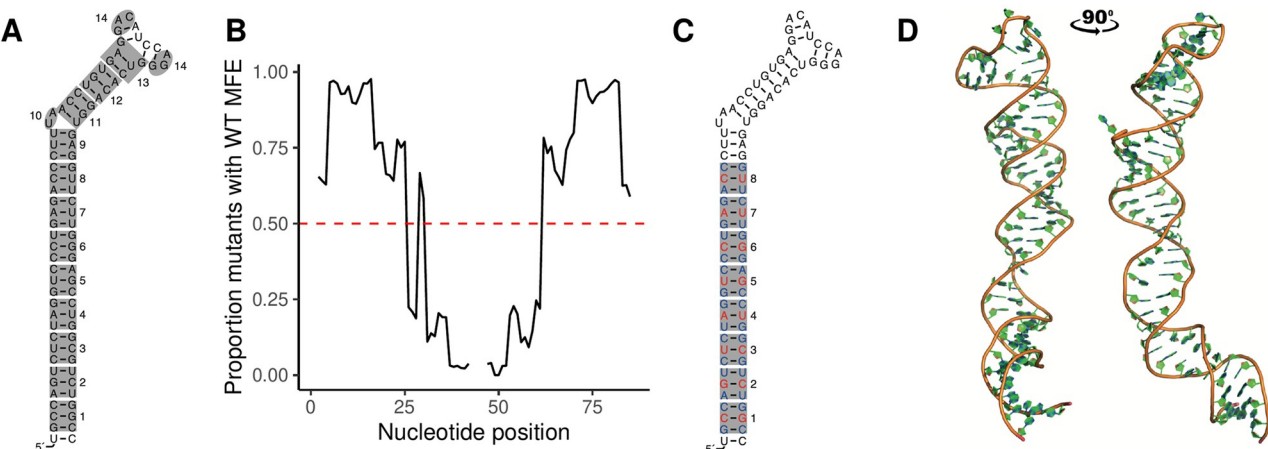

**Fig 2. miR-125a 2D MFE structure, mutation boxes, hard benchmark sets and 3D structure.** A) The 14 boxes that were each exhaustively mutated in the Bartel study, shown on the MC-Fold predicted WT MFE structure. B) Proportions of sequence variants containing each position which adopt the WT MFE structure. C) Positions in the hard benchmark testing set shown in red, and positions in the training set in blue. D) The medoid 3D miR-125a structure from the 67 structures predicted by MC-Sym.

which ensures all mutated positions in the testing set were never mutated in the training set, and an inverted benchmark which minimizes sequence redundancy.

In the hard benchmark, the middle base pair of mutated boxes 1–8 was reserved for the testing set while all sequence variants affecting only the bottom and top base pairs of every box were selected for the training set (illustrated in Fig 2C). This left us with 1849 sequence variants in the training set and 116 in the testing set. The inverted benchmark was designed by taking all WT MFE-preserving variants containing 1 or 2 mutations as the training set, and the rest (3 or more mutations) as the testing set. We call this benchmark inverted since it contains 1094 variants in the training set and 25 866 variants in the testing set. This approach minimizes the ability of sequence-based models to "memorize" maturation efficiency from patterns of several mutations happening in several variants. For example, in a specific box, a specific triplet of mutations will happen in as many as 64 variants across the whole dataset. The effect of that specific triplet could be complex in nature (dynamical, structural, chemical, etc.) and still be captured by a sequence-based model. Hence, the limitation of the training set to variants with at most 2 mutated positions eliminates this artifactual feature while allowing the learning of pure sequence features. For each benchmark, a pair of CSV files containing the selected variants along with their measured maturation efficiency is available. For the hard benchmark, S1 File contains the training set and S2 File the testing set. For the inverted benchmark, S3 File contains the training set and S4 File the testing set.

To generate a 3D model of pri-miR-125a, MC-Sym [36] was run using the WT sequence and 2D MFE structure as input, with default parameters. This generated 67 predicted 3D structures and the medoid structure (shown in Fig 2D) was selected as the template on which all mutations were performed *in silico* using the ModeRNA software [40].

## Multivariate linear regression

After running the ENMs on the pri-miR-125a 3D models of the variants, each position in the Entropic Signature was standardized and used as an input variable to train a LASSO linear regression model [48] to predict the miR-125a maturation efficiency. The inclusion of the standardized MC-Fold enthalpy of folding in the model was also tested. We restricted

ourselves to a linear regression model to allow the pinpointing of important regions in the pri-miR-125a structure, and the LASSO model has the added benefit of selecting important features by driving most coefficients to zero when there is redundancy/covariance between the input variables.

As a sanity check to ensure a sequence-only model cannot learn anything from our hard benchmark, we also tested a sequence vector model. The vector is composed of 4N positions for an RNA sequence of length N where each position can only take values of 1 or -1. For each block of four positions, only one takes the value 1: the first if the nucleotide at the corresponding position in the sequence is an A, the second for a C, the third for a G and the last for a U. In theory, such a model could learn sequence features from the inverted benchmark but should not be able to capture anything from the hard benchmark as the variables that change in the test set were constant across the entire training set.

## Computational cost

To assess the computational cost of the tested ENMs, we ran all models on the RNA chain from the Thermus thermophilus 30S ribosomal subunit [49]. We chose such a large structure of 1514 nucleotides (4541 beads) to highlight the asymptotic behaviour of the tested models, as the building and diagonalizing of the Hessian take time proportional to the cube of the number of beads in the system. The use of a large structure makes operations of linear time like the parsing of the PDB file and the assignment of the beads take a smaller proportion of compute time and allows for a more direct comparison of the costs associated with each different potential. We ran PD-ANM and Cut-ANM with the best parameters found within the B-factors prediction benchmark. In addition, we ran Cut-ANM at 5 Å distance cutoff to see the effect of having a sparser Hessian to diagonalize. As everywhere else throughout the present study, ENCoM was used without modifying the default parameters. The hardware used for these tests is the AMD Rome 7532 (2.40 GHz) as part of Calcul Québec's Narval supercomputer, restricted to 1 CPU with the Slurm shcheduler. We repeated the test 10 times for each model to account for variability in the disk read times and other potential discrepancies. We also ran the same protocol using our 3D model of miR-125a (86 nucleotides, 258 beads), this time building the models and Entropic Signature 100 times per job and reporting the mean compute time for 10 repetitions. This second test illustrates the computational cost of a more modestly sized structure, which is also the focus of our high-throughput mutational analysis.

## Results

### Experimental B-factors prediction

X-ray crystallography gives rise to experimental temperature factors commonly called B-factors, which measure how much each atom oscillates around its equilibrium position in the crystal. The correlation between these fluctuations and those predicted by ENMs is a common benchmark, which we have performed here on X-ray RNA structures of 2.5 Å resolution or less. The clustering step left us with 38 high-resolution structures divided in 34 sequence clusters. To predict the B-factors, we compared the classical mean-square fluctuations (MSF) with the Entropic Signature we introduce (see Materials and methods). Fig 3A shows the Pearson correlation between the predicted and experimental B-factors for the three tested models as a function of the $\beta$ thermodynamic scaling factor. For Cut-ANM and PD-ANM, we also explored their respective parameters extensively and use the ones which give the highest mean correlation (power dependency of 7 for PD-ANM and distance cutoff of 10 Å for Cut-ANM). Fig B in S1 Text shows the details of the parameter exploration, highlighting the discovery of a clear maximum performance. As stated before, we did not test any other ENCoM parameters

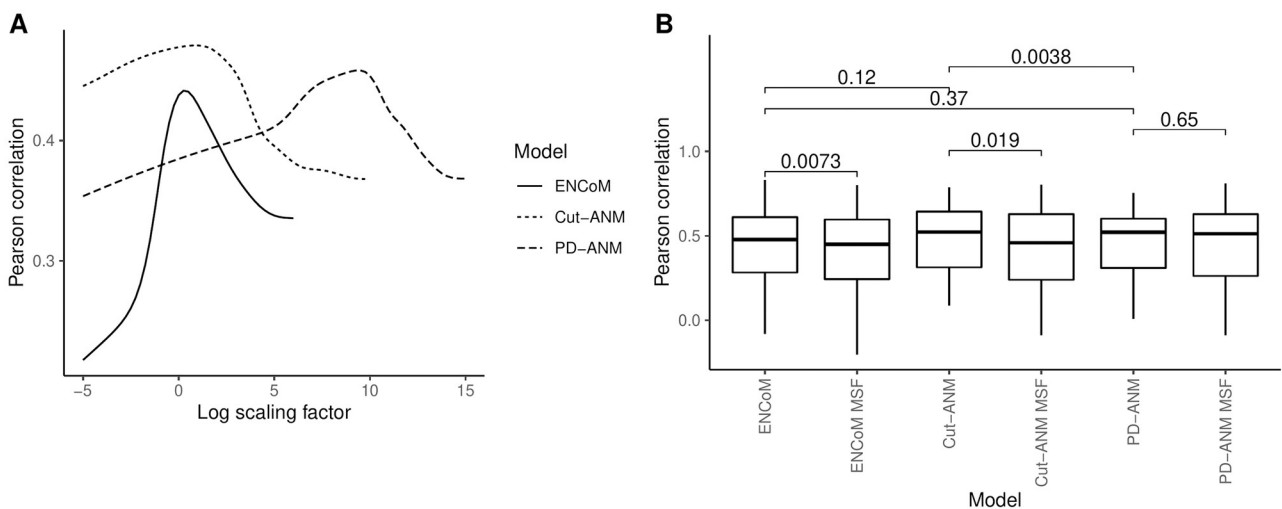

**Fig 3. Pearson correlation coefficient between predicted and experimental B-factors forENCoM and ANM.** A) The mean Pearson correlation is shown as a function of the $\beta$ scaling factor for the three ENMs. B) The Entropic Signature using the best scaling factor for each model is compared to the mean square fluctuations (MSF). The p-values from paired Wilcoxon signed-rank tests are shown for pairs of the two types of predictions for every model and for every pair of models, only for the Entropic Signature predictions since they outperform the MSF.

than the ones already published for proteins. Fig 3B shows the B-factor benchmark in boxplot format, comparing the performance of the three ENMs using either the Entropic Signature (ES) or the classical MSF as a prediction of experimental B-factors. A paired Wilcoxon signed-rank test [50] is performed between the two different forms of prediction for each model. Both ENCoM and Cut-ANM benefit significantly from the ES predictions as opposed to MSF (p = 0.0074 and p = 0.019, respectively), while the benefits for PD-ANM do not reach statistical significance on this dataset (p = 0.65). Since all models perform better using the ES, paired Wilcoxon signed-rank tests are performed between all pairs of ENMs exclusively for the ES predictions. Only the pairwise comparison between Cut-ANM and PD-ANM is statistically significant at p = 0.0038 despite ENCoM having the worst average performance of the three ENMs. The reason for this is that the two flavors of ANM have a similar performance profile across the 34 sequence clusters, while ENCoM tends to perform well where they don't and vice versa. This pattern is apparent when examining the list of correlations for every of the 38 structures shown in Table J in S1 Text. For the MSF predictions, the mean correlations are 0.42 for ENCoM, 0.43 for Cut-ANM and 0.45 for PD-ANM while for the ES predictions, they are 0.44 for ENCoM, 0.48 for Cut-ANM and 0.46 for PD-ANM. These values seem a little low when compared to the correlations previously obtained with ENCoM and other tools applied to proteins [12], all between 0.54 and 0.60. In addition, B-factor correlations of up to around 0.60 have been reported with ANM applied on RNA in a similar fashion as here, with 3 beads per residue, by Zimmermann and Jernigan [7]. However, these studies were all done with only one target B-factor value per nucleotide even when the models used more beads, whereas we correlated the predicted fluctuations at every position in our systems (3 beads per nucleotide).

For three of the 34 sequence clusters used in the experimental B-factors prediction benchmark, both ENCoM and ANM get very low and sometimes negative correlations (shown in Table J in S1 Text, clusters 15, 26, 28). These results could be due to artifacts such as crystal packing or buffer contents, which are not considered by the models, driving the B-factors. If we exclude the cases from these three clusters as experimental flaws and take the mean of the 31 remaining clusters, the correlations we get for the MSF predictions (values in parentheses

are the correlations witout excluding these three clusters) are 0.47 (0.42) for ENCoM, 0.48 (0.43) for Cut-ANM and 0.49 (0.45) for PD-ANM while for the ES predictions, they are 0.49 (0.44) for ENCoM, 0.51 (0.48) for Cut-ANM and 0.49 (0.46) for PD-ANM. These values are closer to what is observed when applying NMA models to proteins [12].

### Prediction of conformational changes

Another source of experimental structural dynamics arising from X-ray crystallography is the crystallization of the same molecule in different conformations due to either varying experimental conditions or the presence of mutations in the sequence of residues. After clustering and applying filters (see Materials and methods), we obtain for this conformational change benchmark 129 structures divided in 29 sequence clusters, forming 240 distinct pairs of conformations with RMSD of at least 2 Å. Each of the clusters contains at least one pair of distinct conformations for a given RNA. We use the 240 pairs of conformations to test the models' ability to predict conformational change. This ability is in many ways more relevant to a biological context, as the conformational variation resulting from different experimental conditions happens on much longer timescales than B-factors and can have functional significance [16]. Fig 4 presents the average cumulative overlap from pairs of different conformations, computed from a single mode up to 10% of the total number of normal modes. The idea behind this as opposed to the more frequent use of a fixed number of modes is that since the number of modes grows linearly with the number of beads in the system ($3N − 6$ nontrivial modes for N beads), it is more logical to use a linear proportion. Fig 4B shows the results as boxplots when 5% of the normal modes are used. The average number of modes selected this way for the current benchmark is 28 (data shown in Fig E in S1 Text), which falls slightly outside the range of 3 to 20 modes tested by Zimmermann *et al.* [7]. In a real-case scenario, limiting the analysis to a small number of modes is mandatory because of the exponential growth of the conformational space they represent, however relevant modes could still be selected among a fixed linear proportion of the total.

At 5% of nontrivial normal modes, the mean cumulative overlap is 0.73 for ENCoM, 0.67 for Cut-ANM and 0.69 for PD-ANM. The p-values from a paired Wilcoxon signed-rank test are shown for each pair of models, illustrating that the advantages ENCoM has over both ANMs and the advantage PD-ANM has over Cut-ANM are very significant. This result is consistent with our previous results benchmarking ENCoM on proteins, where ENCoM performed highest of all the models tested on the conformational change benchmark [12].

Fig 4C shows the cumulative overlap at 5% nontrivial modes as a function of the conformational change RMSD, which ranges from the lower bound of 2 Å up to 26 Å. The structure pairs were first put in bins of 0.1 Å RMSD to enhance visibility. The presence of several high RMSD pairs around 22 Å begged further investigation. Fig 4D illustrates an example of such a high RMSD pair, which happens as an RNA strand switches between a duplex and hairpin conformation. Interestingly, ENCoM has a big performance edge over ANM for such pairs of structures. This big gain can be explained by the greater physical awareness of the ENCoM potential, which only restricts motion between beads close in sequence or which form favorable atomic contacts. In contrast, ANM connects vast numbers of beads, thus biasing the conformational landscape by maintaining unrealistic interactions. The detailed list of

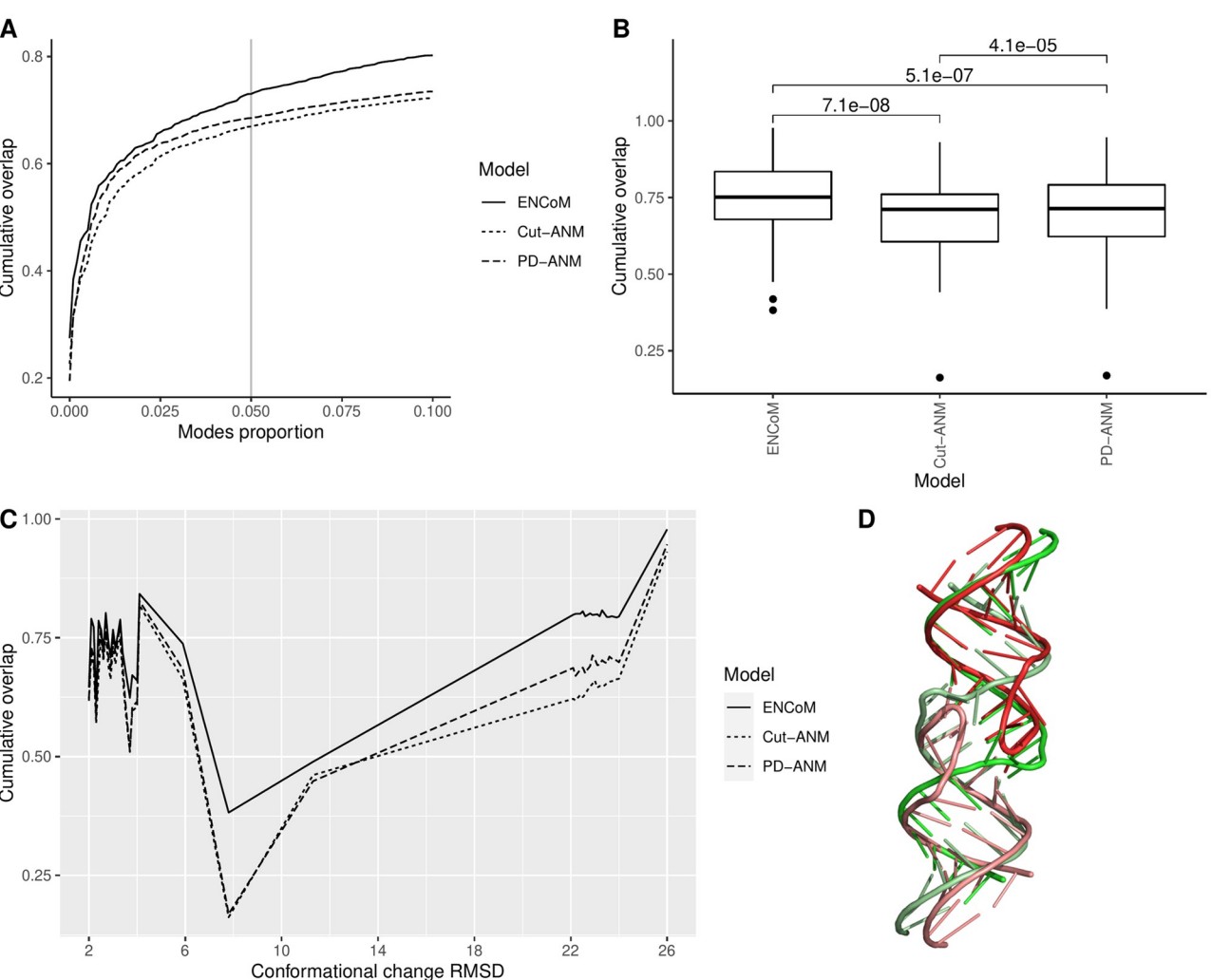

**Fig 4. Cumulative overlap between normal modes and conformational changes from X-ray crystallography experiments.** A) Mean cumulative overlap as a function of the proportion of the nontrivial normal modes used. B) Mean cumulative overlap at 5% nontrivial normal modes used, with p-values from paired Wilcoxon signed-rank tests for every pair of models. C) Mean cumulative overlap at 5% nontrivial modes, as a function of the conformational change RMSD, in bins of 0.1 RMSD. D) An illustrative example from the high RMSD pairs, PDB codes 2B8R (red) and 3FAR (green), with 22.8 RMSD. One strand from each structure is colored paler than the other.

performance at 5% nontrivial modes for each model, PDB codes and RMSD for every pair in the conformational benchmark can be found in Table K in S1 Text.

## Capturing NMR ensemble variance

Perhaps the most reliable source of experimental information regarding the 3D structural dynamics of RNA comes from solution NMR experiments. In contrast to X-ray crystallography, NMR experiments give information about the conformational space of the studied molecule over vast timescales as part of a single experiment. The resulting ensemble contains models which taken together explain the experimental obsevables. The motions apparent within these structural ensembles can inform us on the ability of NMA models to recreate the conformational space of a molecule from its lowest energy conformation. After filtering and clustering (see Materials and methods), we are left with 313 solution NMR ensembles in 271

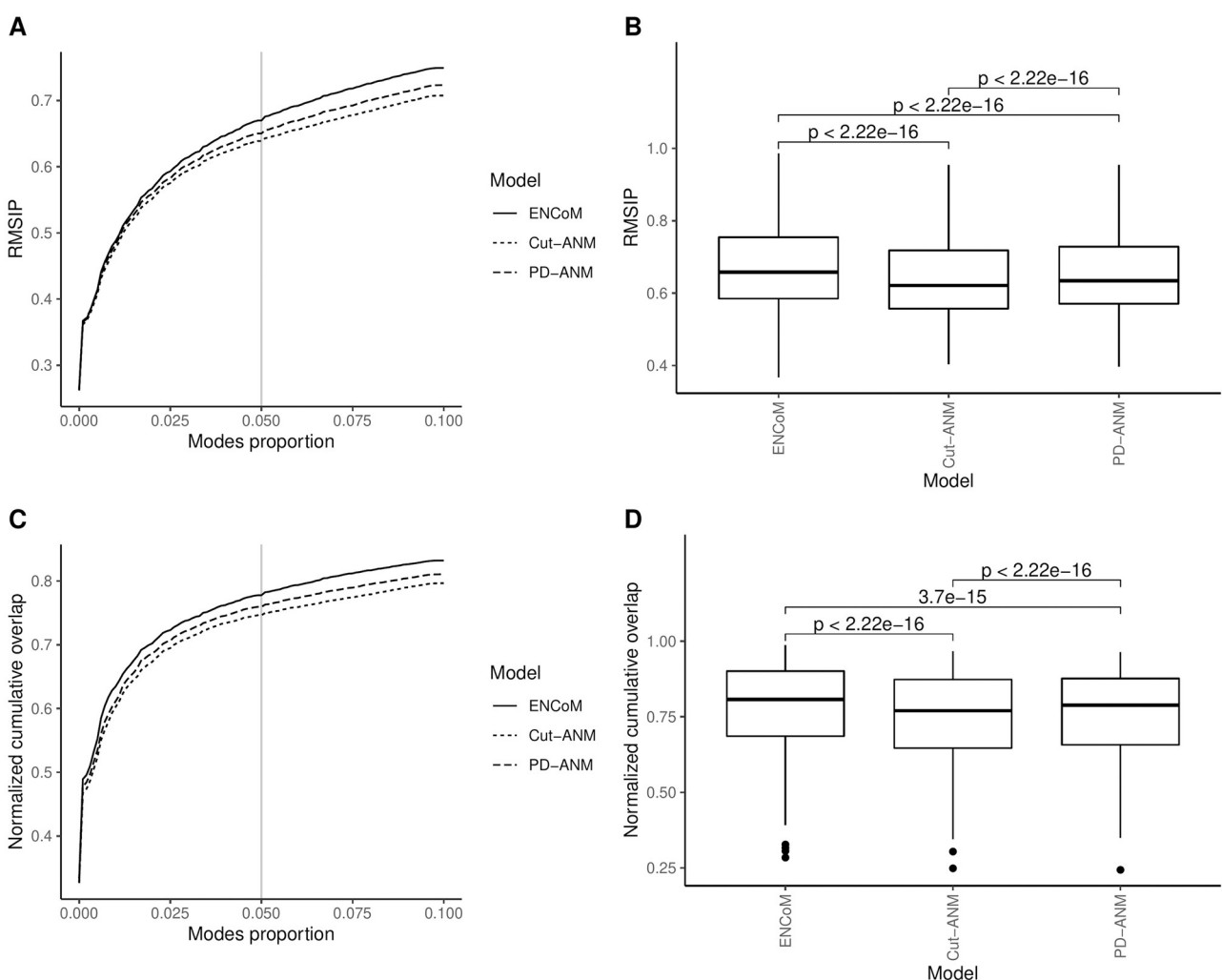

**Fig 5. RMSIP and NCO between normal modes and principal components from NMR ensembles.** A) Root mean square inner product (RMSIP) between a proportion of the nontrivial normal modes and the principal components accounting for at least 99% of the variance apparent within the NMR ensemble. B) RMSIP at 5% nontrivial normal modes, with p-values from paired Wilcoxon signed-rank tests for every pair of models. C) Normalized cumulative overlap (NCO) between a proportion of the nontrivial normal modes and the principal components accounting for at least 99% of the variance apparent within the NMR ensemble. D) NCO at 5% nontrivial normal modes, with p-values from paired Wilcoxon signed-rank tests for every pair of models.

sequence clusters. We test each ENM's ability to capture the internal motions apparent from these ensembles by computing the RMSIP and NCO between PCs accounting for 99% of the ensemble variance and a number of the lowest frequency nontrivial modes computed the ensemble's equilibrium structure. We use from 1 mode up to 10% of the nontrivial normal modes and report the performance across this range in Fig 5, panels A and C (RMSIP and NCO, respectively). Panels B and D present the performance at 5% nontrivial normal modes, which corresponds to an average number of 14 normal modes (see Fig E in S1 Text).

The average RMSIP at 5% normal modes is 0.67 for ENCoM, 0.64 for Cut-ANM and 0.65 for PD-ANM. The average NCO, also at 5% normal modes, is 0.78 for ENCoM, 0.75 for Cut-ANM and 0.76 for PD-ANM. Again, the p-values from Wilcoxon signed-rank tests are shown for every pair of models. For the two metrics, the difference in performance is highly significant between each pair of models ($p < 10^{-14}$). Thus, in addition to the prediction of

conformational change, ENCoM has the advantage on the prediction of ensemble variance, seemingly confirming that the motions it predicts happen on longer timescales than B-factors. The detailed performances, again at 5% nontrivial modes, are listed in Table L in S1 Text.

### Predicting miR-125a maturation efficiency

To ensure a stringent assessment of whether the ENMs tested can capture the influence of 3D structural dynamics on miR-125a maturation efficiency beyond sequence features, we put together a training set of 1849 sequence variants and a testing set of 116 which have no mutated position in common. Fig 6A shows the predictive $R^2$ from LASSO models trained on the training set and applied to the testing set. For the three ENMs, the Entropic Signature (ES, see Materials and methods) is computed for 41 values of the $\beta$ scaling factor centered around the one which gave the best performance on the B-factors benchmark ($e^{0.25}$ for ENCoM, $e^1$ for Cut-ANM, $e^{9.5}$ for PD-ANM), varied logarithmically in increments of 0.25. In addition, the regularization strength $\lambda$ of the LASSO model is tested in $\log_2$ increments from $2^{-15}$ to 1. We also tested a sequence vector (see Materials and methods) to validate that the hard benchmark truly prohibits the learning of sequence features. Apart from these four models on their own, the addition of the MC-Fold enthalpy of folding to the LASSO model is tested and the results are shown on the bottom row of Fig 6A. Interestingly, ENCoM is the only ENM for which the ES alone allows the LASSO model to have a positive predictive $R^2$. The maximum value of 0.11 is reached at $\beta = e^{0.25}$, which is the center value from the swept $\beta$ parameters and the one which gave the best performance in the B-factors prediction benchmark. The MC-Fold enthalpy of folding alone also reaches a modest positive predictive $R^2$ of 0.07. As expected, the sequence vector ends up being a constant predictor and its combination with the MC-Fold energy produces the exact same cloud of points as MC-Fold alone. On their own, Cut-ANM and PD-ANM fail to capture any of the variance in the testing set as illustrated by their predictive $R^2$ values being zero or less for all parameters scanned. This result was expected as these models are completely agnostic to the chemical nature of the nucleobases. The combination of ENCoM with MC-Fold energy seems synergistic, giving rise to a maximal predictive $R^2$ of 0.22. To assess the significance of this value, we simulated the combination of the MC-Fold alone predictions with random predictions generated as a normal distribution centered around the training set mean, with half its standard deviation. The results of this simulation are shown in Fig F in S1 Text. The p-value for the combination of the different models with MC-Fold being significantly better than just MC-Fold combined with noise are shown in the bottom row of Fig 6B. The combination of the ENCoM ES with MC-Fold leads to a highly significant improvement with $p < 0.0001$.

Fig 7 gives the performance of every model on the inverted benchmark, which contains sequence redundancy between training and testing sets. As expected, the sequence model alone captures some, although slight, signal at 0.06 predictive $R^2$. Perhaps surprising is the performance of ANM alone, in particular PD-ANM, at 0.20 predictive $R^2$. Still, ENCoM maintains a robust advantage at 0.34 predictive $R^2$. Moreover, the combination of all models with the MC-Fold enthalpy is highly beneficial, reaching 0.56 predictive $R^2$ for the sequence vector, 0.59 for both flavors of ANM and 0.66 for ENCoM. Thus, the MC-Fold-ANM combination captures an additional 3% of variance compared to the MC-Fold-Sequence combination, while ENCoM-MC-Fold captures an additional 10%. These results do not imply ANM exhibits chemically relevant sequence sensitivity. Instead, the linear model is able to learn the sequence from specific patterns in the Entropic Signatures arising from the slight changes in the cartesian coordinates of the nucleobase beads (the C2 atom for a given nucleobase has unique

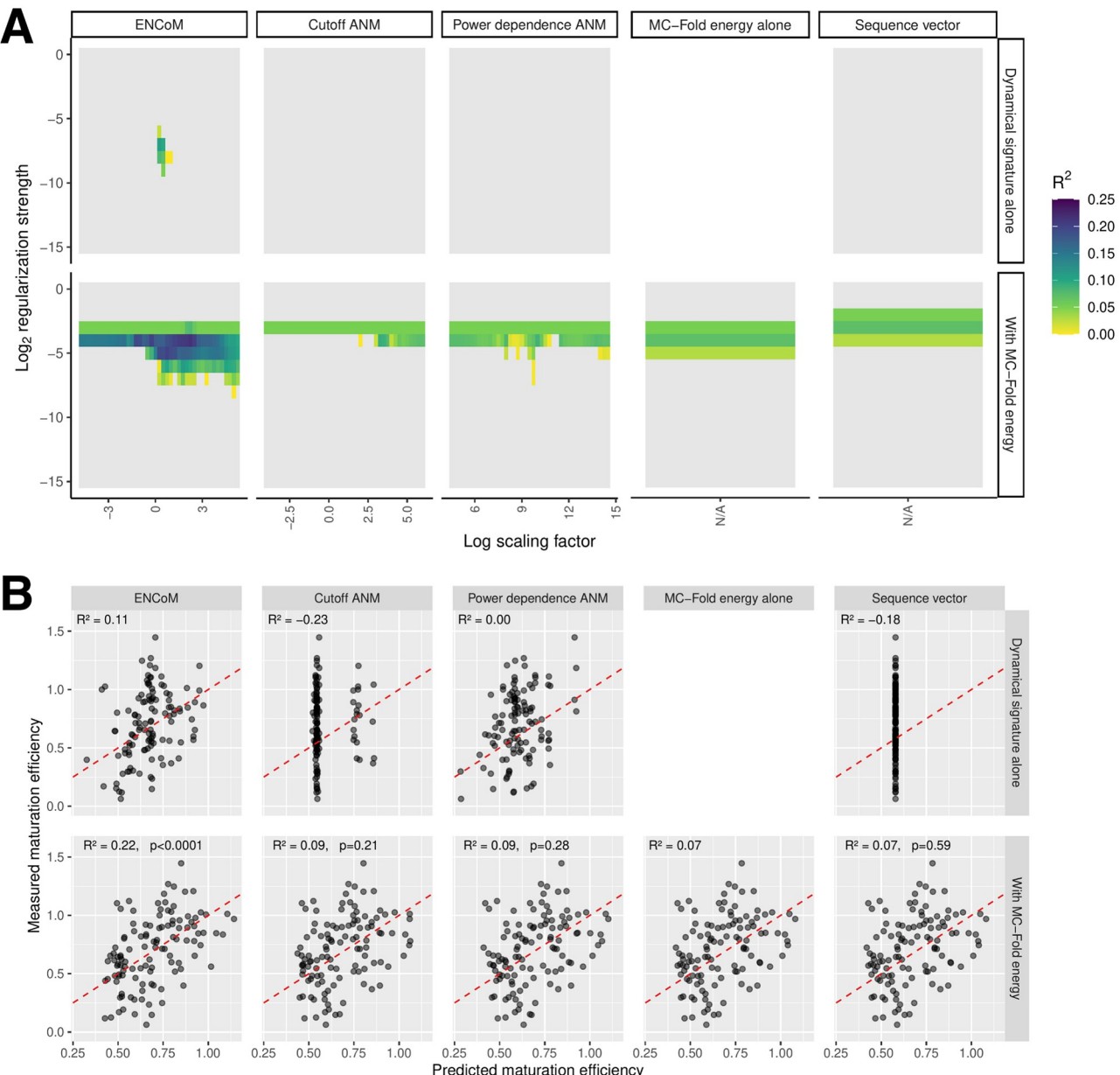

**Fig 6. Performance of LASSO linear regression on the hard benchmark.** A) Predictive $R^2$ for each model alone or in combination with the MC-Fold enthalpy of folding. For the three ENMs, scaling factors for the dynamical signature were explored around the value which gave the best respective performance in the B-factors benchmark. Predictive $R^2$ values below zero are shown in gray. B) Performance on the test set for the best combination of parameters for every model alone or in tandem with MC-Fold. The bottom row shows the p-value from a simulation combining gaussian noise with the MC-Fold prediction, which corresponds to the probability of these predictive $R^2$ values arising by chance from a pure noise model. The dotted red lines show $x = y$.

coordiates for each possible mutation). ANM's lack of chemical sensitivity is evident from Fig 1, panels C and D, as already discussed.

The best predictive performance on our hard benchmark of miR-125a maturation efficiency is reached by the combination of the ENCoM ES at $\beta = e^{2.25}$ with the MC-Fold enthalpy of folding, fed into a LASSO linear regression model with regularization strength $\lambda = 2^{-4}$. The coefficients of the resulting model are shown in graph format in Fig 8A and mapped back on

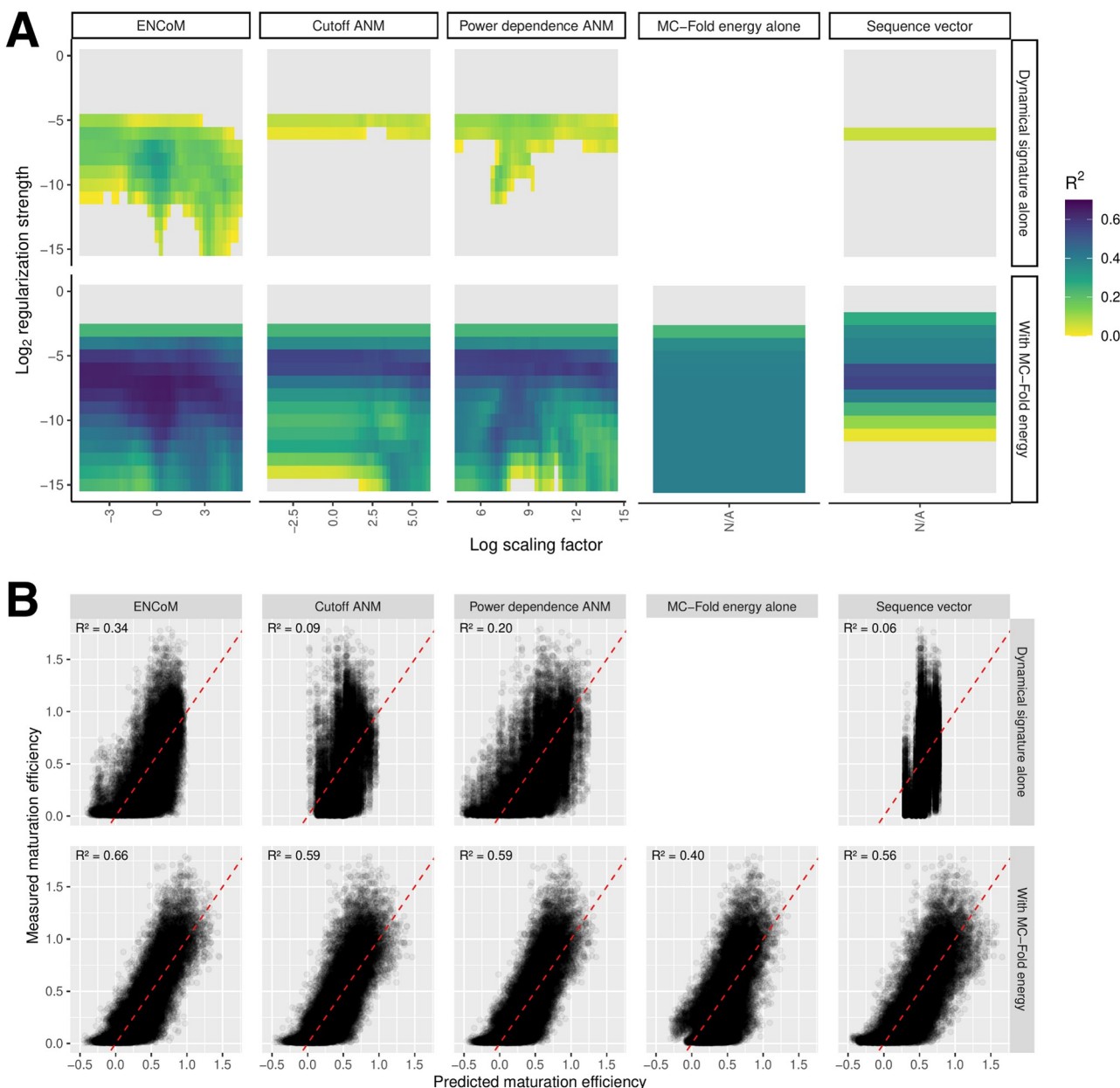

**Fig 7. Performance of LASSO linear regression on the inverted benchmark.** A) Predictive $R^2$ for each model alone or in combination with the MC-Fold enthalpy of folding, as in Fig 6. B) Performance on the test set for the best combination of parameters for every model alone or in tandem with MC-Fold.

the 2D structure of pri-miR-125a in Fig 8B. The highest remaining coefficient from the ENCoM ES is a positive coefficient at the phosphate bead of nucleotide 78. This pattern corresponds to a motif previously identified by Fang *et al.* as the mismatched GHG motif [26].

Strikingly, the mismatched position is included in the testing set, meaning that the model learns the importance of flexibility at a position never mutated in the training set. All variables in the model are standardized so the coefficients can be directly compared as indicators of the variables' relative importances. With this model, the MC-Fold energy has a coefficient of -0.19

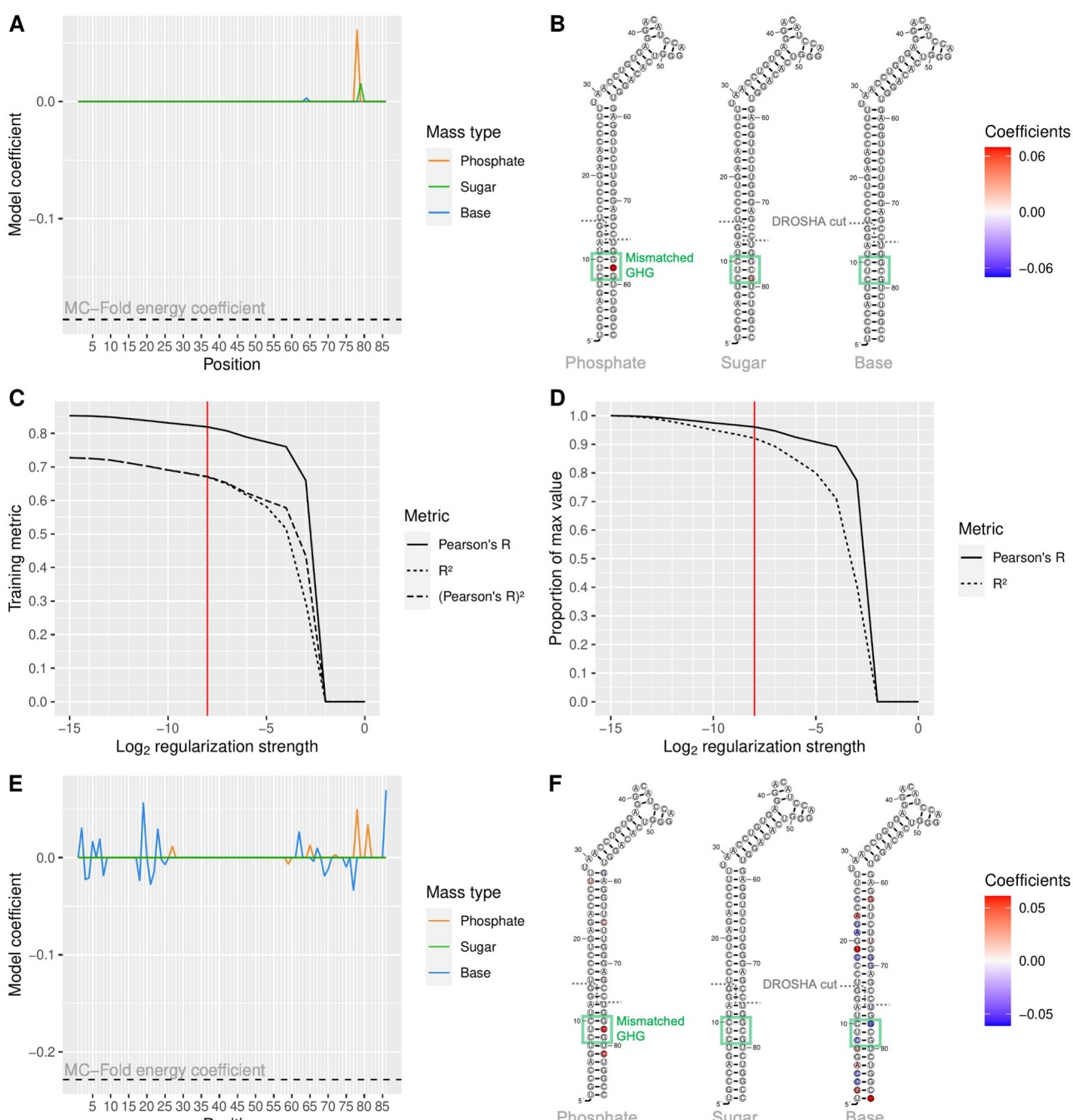

**Fig 8. Coefficients of LASSO regression models.** A) Coefficients of the combined ENCoM and MC-Fold model trained on the hard training set with $\beta = e^{2.25}$ and $\lambda = 2^{-4}$. B) ENCoM Entropic Signature coefficients mapped on the 2D structure of pri-miR-125a. The DROSHA cut site and mismatched GHG motif are identified on the structures. C) Training metrics from training the ENCoM-MC-Fold combination at $\beta = e^{2.25}$ on the whole set of 26 960 sequence variants selected for this study. Pearson's R, $R^2$ and the square of Pearson's R are plotted in relation to regularization strength. A red vertical line shows the last regularization strength before $R^2$ starts to diverge from the square of Pearson's R. D) Pearson's R and $R^2$ training performance expressed as a proportion of maximal performance, as a function of LASSO regularization strength. E)-F) Same as A)-B), with the LASSO model trained on all 26 960 sequences and $\lambda = 2^{-8}$.

while the sum of ENCoM coefficients is 0.08. These coefficients point to the model favoring a low enthalpy of folding but an entropically flexible structure at the same time, especially at the GHG motif. All the ENCoM coefficients are positive with these parameters so the sum of absolute coefficients is also 0.08, having roughly half the importance of the MC-Fold energy.

The most generalizable learning from the hard benchmark happened with the scaling factor $\beta = e^{2.25}$. This same scaling factor leads to a predictive $R^2$ value of 0.65 for the MC-Fold-ENCoM combination on the inverted benchmark, comparing favorably to the 0.66 maximum reached. Thus, using this exact $\beta$ value, we trained another round of LASSO models, this time using the whole set of 26 960 mutant sequences from boxes 1–8 with WT MFE as the training set. Fig 8C illustrates the training performance of the model as measured by Pearson's R linear correlation, $R^2$ and the square of Pearson's R. For high enough regularization strength, $R^2$ dips below the square of Pearson's R, which points to the model being underpowered due to the high regularization. We thus chose $\lambda = 2^{-8}$ as a good tradeoff between the reduction of model variance and the maintaining of good performance. In Fig 8D, we can see that training $R^2$ stays above 90% of its maximal value and Pearson's R above 95% at that regularization strength. The coefficients from the LASSO model trained on all sequences with $\beta = e^{2.25}$, $\lambda = 2^{-8}$ are shown in panels E and F in a similar fashion as in panels A and B. The lower regularization strength allows interesting patterns to emerge at nucleobase beads throughout the structure while still being high enough to drive most backbone coefficients to zero. Here again, the most important backbone feature is flexibility at the GHG mismatch motif. At this regularization strength, the MC-Fold energy coefficient is -0.23, the sum of ENCoM coefficients is 0.15 and the sum of absolute ENCoM coefficients is 0.59. The overall trend of low enthalpy and high vibrational entropy being favorable for processing is still present, but the ENCoM coefficients have almost three times the relative importance of the MC-Fold enthalpy.

## Computational cost

The results of the computational cost analysis are given in Fig G in S1 Text. For the Thermus thermophilus 30S ribosomal subunit, on average, ENCoM takes 26.5 minutes, Cut-ANM takes 19.2 and 18.8 minutes for the 18 and 5 Å cutoff values respectively and PD-ANM takes 22.7 minutes. For miR-125a, on average for building one model and computing one Entropic Signature, ENCoM takes 3.33 seconds, Cut-ANM takes 1.63 seconds while the 5 Å version takes 1.56 seconds and PD-ANM takes 2.30 seconds. These results illustrate that for large structures, the difference in CPU cost between the models becomes smaller as more time is spent on solving the Hessian matrix, while for small structures the difference is much more significant since the building of the Hessian takes more time for more complex (ENCoM) or more interconnected (PD-ANM) potentials. Cut-ANM is faster at a lower distance cutoff because it leads to a sparser Hessian which is in turn faster to solve.

## Discussion

### Overall performance and parameterization

In the present work, we have extended ENCoM to RNA and shown gains in performance when compared to ANM for the prediction of large conformational changes, at the expense of a small loss in the ability to predict B-factors from crystallographic experiments. Namely, ENCoM shows an advantage when it comes to conformational change prediction and NMR ensemble variance prediction. As stated, we believe that the performance of coarse-grained normal mode analysis models in predicting biologically relevant motions is explained by the geometric hypothesis: they capture something fundamental about the shape of the equilibrium structure. In that sense, STeM and ENCoM add relevant layers of connectivity information to

the geometric description, thus resulting in performance gains. ENCoM further includes sequence sensitivity from atomic surfaces in contact, mapped back to the appropriate coarse-grained beads. It thus follows from this reasoning that parameter exploration was unnecessary in order to reap the benefits of ENCoM on RNA structures. We remind the reader that we extensively tested the single parameter for both versions of ANM and used the parameters leading to maximal performance, yet still found better performance from ENCoM on relevant benchmarks. While the ENCoM model is very robust in performance across vast parameter space, it would be interesting to explore this space in relation with performance on RNA structures. We leave this exploration for future work as our goal would be to find parameters optimizing performance on RNA, proteins and their complexes, a task which is beyond the scope of the present work. For now, as stated, the default ENCoM parameters provide a measurable advantage over other ENMs on RNA structures.

Our main interest in extending ENCoM to RNA was to use it to study the effect of mutations as we have already done in the case of proteins [13, 15]. The present work illustrates that ENCoM indeed extends this capability to RNA, which is not the case of the sequence agnostic ANM model. In our opinion, our findings make ENCoM the preferred model wherever large-scale, low computational cost conformational dynamics are wanted.

### Artifacts from crystallographic B-factors

It might be worthwhile to investigate in more details the nature of the quasi-null correlations obtained for three out of the 34 sequence clusters used for the B-factors prediction benchmark to see if some factor in the experimental conditions or the nature of the RNAs could explain the discrepancy. However, in the case of ENCoM, we are more interested in predicting conformational changes than B-factors, and we have previously shown that parameter optimization for our model offers a trade-off between conformational space and B-factors prediction [12]. Furthermore, B-factors have been shown to adopt biphasic behavior with a different slope of variation for cryogenic versus standard temperatures [51]. Since most X-ray crystallography experiments are conducted at cryogenic temperatures, their B-factors measurements are thus less relevant to biological function. B-factors can also arise from crystal packing and rigid body motions which are irrelevant in a biological context [52].

### Large-scale conformational change prediction

When it comes to the prediction of large-scale, slow timescale conformational change, whether from pairs of X-ray structures in different conformations or from NMR ensembles, ENCoM has a clear advantage over both Cut-ANM and PD-ANM. This better performance comes as no surprise in view of our previous studies [12] and reinforces the idea that moving the coarse-grained ENMs in the direction of more physical realism is beneficial. One innovative further development that could be added to ENCoM is the inclusion of an electrostatic term. In the case of RNA especially, electrostatic interactions are of primordial importance and capturing their relative strengths within the model would most probably further increase its accuracy. The simplest way to achieve this would be to change the $\beta_{ij}$ term to make it depend on electrostatic calculations on the input structure. This addition would up the computational cost of the model considerably and require careful benchmarking, yet could provide important performance gains.

### Comparison to MD simulations

While it is hard to imagine running MD simulations on the whole set of miR-125a variants, we have done so on the 16 possible variants at the base pair of the human SNP (G22-C65).

Accumulating 3 replicate trajectories of over 100ns for each variant represented a computational cost of approximately 40 core-years. For comparison, running ENCoM on these 16 variants takes less than a minute. The fluctuation profiles from the MD simulations did not lead to significant correlation with experimental data, probably because of insufficient simulation length. We present the detailed protocol and results in S1 Text.

## Predicting the effect of mutations from Entropic Signatures

All coarse-grained ENMs have in common the pseudo-physical nature of their potentials. Their eigenfrequencies do not have defined units, and the same goes for the vibrational entropy values we can compute from them. However, the relative ranking of eigenvalues is very relevant to biological function as the slowest normal modes capture the most global, cooperative motions which are often critical for macromolecular function. In this work we introduced the Entropic Signature to reduce the conformational space from all normal modes to a vector of fluctuations at every bead in the system. In contrast to the classically used mean square fluctuations, the addition of a thermodynamic scaling factor as part of the vibrational entropy calculation allows for varying the relative contributions of slow and fast motions to the dynamical signature. This is of utmost importance considering the pseudo-physical nature of the models, as parameters such as temperature and physical constants do not have a clear meaning for the coarse-grained ENMs.

The hard miR-125a maturation efficiency benchmark illustrates this importance of fitting the scaling factor to the studied system. For the ENCoM dynamical signature alone, only a small range of $\beta$ values gave rise to positive predictive $R^2$. At the same time, the best $\beta$ values are close to the value selected from the B-factors prediction benchmark. Thus, it seems that a small range of values might be appropriate for a large range of different molecules.

To our knowledge, this is the first time 3D structural dynamical information has been used in such a way to train multivariate linear models in predicting the effect of mutations. The method has the advantage of a powerful statistical model while maintaining the necessary transparency to allow the identification of important regions and patterns directly on the structure. The drawback is the inability of the model to capture complex relationships between the flexibility of different regions, as every input variable is linearly independent.

While the performance of the LASSO model combining ENCoM with the MC-Fold enthalpy on the hard benchmark is modest with a predictive $R^2$ of 0.22, we must keep in mind that perfect performance on this hard test is not expected and might not even be possible. Indeed, it cannot be excluded that sequence features are part of what constitutes the processing signal for miRs, so this benchmark must be seen as a test of generalizability of the model rather than an upper bound on its performance. The performance on the inverted benchmark is much higher, at 0.66 predictive $R^2$. To illustrate the performance ceiling one could expect, Fig C in S1 Text shows the testing performance when randomly splitting the data in 80% for training and 20% for testing. As discussed in Materials and methods, this is a naive approach that leads to vast amounts of sequence redundancy between the two sets. However, the performance of all models is strikingly high, with the combination of ENCoM and MC-Fold reaching 0.75 predictive $R^2$ and the combination of both Cut-ANM and PD-ANM with MC-Fold reaching 0.74. Even the sequence vector reaches 0.71 predictive $R^2$ when combined with the MC-Fold energy. As stated, this test can be seen as an upper bound on the models' performance. It might seem surprising at first that ANM can capture some signal on its own. However, this signal is explained by the fact that the nucleobase beads all have slightly different positioning relative to the backbone. The different positions of the beads thus lead to a sequence signal being present in ANM's conformational space, which the LASSO model can

learn. Performing the same experiment with only one bead per nucleotide, located at the phosphorous atom, leads to the signal completely disappearing for ANM, since the backbone is fixed and thus the dynamical signatures computed for all variants are identical. As for the appearance of a performance ceiling for this 80/20 train/test split, we hypothesize that noise in the experimental data is causing it. One possibility is that since the data is generated by incubating all mutants in the same test tube before sequencing their barcodes to determine processing efficiency [26], it allows for interactions between different molecules to take place, potentially interfering with sequencing accuracy.

## Dynamical features necessary for miR-125a processing

Perhaps the most interesting application of our multivariate linear model applied to ENCoM dynamical signatures is the possibility to map the selected coefficients back to the structure of the studied molecule in order to gain biological insights. In the present case of miR-125a, the selection of the noncanonical UC base pair from the mismatched GHG motif by the LASSO model as the most important feature from the dynamical signature strikes us as very promising in terms of our approach's ability to lead to relevant insights. An important finding from our study is that the model favors sequences which have both low folding energy and high vibrational entropy. This finding is at odds with Fang and Bartel's model of pri-miR processing in which the gain of more rigid base pairs throughout the whole stem is associated with higher processing efficiency, with the only exception being the GHG mismatch. However, they analyzed their data with a position-centric view in which the effect of a mutation was deemed to happen at the mutated position only. Our model captures the effect of a mutation on the flexibility of the whole pri-miR, thus enabling the recognition of much finer dynamical patterns happening away from the site of mutation. In fact, our model does not contradict the hypothesis of more rigid base pairs leading to better processing, as this relationship is captured by the MC-Fold enthalpy of folding (which favors canonical, and GC base pairs even more). However, from an evolutionary point of view, a rigid hairpin of 35 GC basepairs minus one mismatch would probably lead to the DNA sequence of the miR gene folding on itself and wreaking havoc on genomic stability when transcribed and replicated. Thus, the maintaining of enough flexibility in the hairpin structure could be a way to prevent the DNA hairpin from forming while still allowing the RNA structure to exist.

When looking at the coefficients for the nucleobase beads in Fig 8F, one apparent pattern is that negative coefficients tend to be overrepresented on base pairs that are already rigid while the reverse is true for positive coefficients. This pattern also points away from rigidity being highly favored, as if it were the case, we should see GU wobble and noncanonical base pairs colored in blue. Instead, the reverse pattern is apparent, with most negative coefficients appearing at GC base pairs and positive coefficients appearing in places where a mismatch is already present like the UG base pair at position 19. There are even two GC base pairs for which more flexibility is favored by the model: at positions 25 and 2. Our results thus point to very complex dynamical patterns being intimately tied to human miR processing and merit further investigation. As part of future work, the LASSO model trained from the whole set of 26 960 sequence variants seems an ideal candidate to predict maturation efficiencies for novel pri-miR-125a variants.

## Conclusions

We presented a version of ENCoM specifically tailored to work optimally on RNA molecules by allowing the use of three beads per nucleotide in the coarse-graining of the input structure. ENCoM consistently outperforms ANM for large-scale conformational space prediction, in

addition to being parameter-free as far as the user is concerned. However, the main advantage of ENCoM is the possibility to estimate the effect of mutations, which ANM is incapable of due to its sequence agnostic nature. Utilizing high-throughput data on the maturation efficiency of miR-125a sequence variants, we show that ENCoM indeed preserves this ability on RNA molecules by the careful construction of a stringent test from the maturation efficiency dataset. Moreover, the inverted benchmark, which contains a careful amount of sequence redundancy, also highlights ENCoM's ability to capture dynamics-function relationships as a consequence of mutations.

Most importantly, the usage of the ENCoM Entropic Signature inside a multivariate linear model is the first of its kind, to the best of our knowledge. It allows us to point back to regions of biological interest on the structure of pri-miR-125a, namely the mismatched GHG motif. We also challenge the notion that rigid base pairs favor processing at almost every position. Our methodology is easily applicable to proteins, RNAs and their complexes, provided some high-throughput data about the biological property of interest is available for the studied molecule, along with an experimentally solved structure or a good 3D model.

## Supporting information

**S1 Text. Supplementary information.** Fig A. Correlation between miR-125a maturation efficiency and Dynamical Signature distance to WT for both ENCoM and MD simulations. Fig B. Cut-ANM and PD-ANM parameter sweep for the B-factors benchmark. Fig C. Performance of multiple linear regression on an 80–20 train-test split. Fig D. Size distributions of the sequence clusters from the 3 benchmarks. Fig E. Number of nontrivial normal modes at 5% for the overlap and NMR benchmarks. Fig F. Simulation of predictive R-squared from combinations of MC-Fold enthalpy with random noise. Fig G. Computational cost of the ENMs on the Thermus thermophilus 30S ribosomal subunit and on miR-125a. Table H. Atom type assignation of the four standard nucleotides. Table I. PDB codes of the structures used for the three benchmarks. Table J. Performance on individual structures for the B-factors correlation benchmark. Table K. Performance on individual pairs of conformations for the X-ray conformational change benchmark. Table L. Performance on individual NMR ensembles for the ensemble variance benchmark. Table M. MD trajectories of miR-125a variants. Table N. ENCoM interaction strength for different base pairs.
(PDF)

**S1 File. Training set for the hard benchmark.**
(CSV)

**S2 File. Testing set for the hard benchmark.**
(CSV)

**S3 File. Training set for the inverted benchmark.**
(CSV)

**S4 File. Testing set for the inverted benchmark.**
(CSV)

## Acknowledgments

We thank Marie Papineau, Frédéric Mailhot and Etienne Richan for their careful reading of the manuscript.

## Author Contributions

**Conceptualization:** Olivier Mailhot, Vincent Frappier, François Major, Rafael J. Najmanovich.

**Data curation:** Olivier Mailhot.

**Funding acquisition:** François Major, Rafael J. Najmanovich.

**Investigation:** Olivier Mailhot.

**Methodology:** Olivier Mailhot, Vincent Frappier, Rafael J. Najmanovich.

**Project administration:** Rafael J. Najmanovich.

**Software:** Olivier Mailhot.

**Supervision:** François Major, Rafael J. Najmanovich.

**Writing – original draft:** Olivier Mailhot, Rafael J. Najmanovich.

**Writing – review & editing:** Olivier Mailhot, François Major, Rafael J. Najmanovich.

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
