## [Decision Letter · Decision Letter 0]

21 Jul 2022

Dear Dr. Najmanovich,

Thank you very much for submitting your manuscript "Sequence-sensitive elastic network captures dynamical features necessary for miR-125a maturation" for consideration at PLOS Computational Biology.

As with all papers reviewed by the journal, your manuscript was reviewed by members of the editorial board and by several independent reviewers. In light of the reviews (below this email), we would like to invite the resubmission of a significantly-revised version that takes into account the reviewers' comments.

In particular, we do think a comparison with MD-simulations is necessary (see reviewer comments)

We cannot make any decision about publication until we have seen the revised manuscript and your response to the reviewers' comments. Your revised manuscript is also likely to be sent to reviewers for further evaluation.

Sincerely,

Shi-Jie Chen

Associate Editor

PLOS Computational Biology

Arne Elofsson

Deputy Editor

PLOS Computational Biology

Reviewer's Responses to Questions

**Comments to the Authors:**

Reviewer #1: The present work offers an interesting coarse grained ENM-based approach to approximate the local dynamics of RNAs. This approach is demonstrated to be sufficiently cheap for large-scale applications while able to predict sequence-specific properties. Using this approach, the authors sucessfully related the dynamics of pri-miR-125a variants to its maturation efficiency. Yet this exciting achievement appears buried in the details and not stressed enough. Moreover, to make this manuscript more convincing, extra evidences shall be provided.

Although molecular dynamics simulations could be too expensive for large-scale validations, a comparison and this approach and MD simulations is still necessary, at least for the wild-type miR-125a and one/two variant sequences. This won't be super costly, since the system is not huge and only local dynamics around the medoid structure of pri-miR-125a need to be sampled. Such comparison not only adds more credit to ENCoM but may also offer clues on future improvements of including electro-static terms.

A related interesting question could be at what temperature shall the MD be performed and what is the relation between this temperature with the parameter \\beta and the entropy of the molecule. Some discussion on this will be helpful.

"The pinpointing of important regions in the pri-miR-125a" shall be stressed more. This could be done by (1) starting a new paragraph at Line 484 from "Strikingly, ..."; (2) Modify Figure 7 so that the GHG motif can stand out from the rest.

Other than these, I find the manuscript clearly written and scientifically sound.

Reviewer #2: In their manuscript, Maillot et al introduce an RNA-version of an elastic network model (ENCoM) originally developed for proteins.

They evaluate the accuracy of the model in predicting B-factors and motions as described by NMR bundles.

Furthermore, they train a regularised linear model to predict maturation efficiency of a miRNA using as input features dynamical information predicted from the elastic network model.

While some of the aspects of this work are innovative, I have a number of concerns.

1. If I understand correctly, the ENCoM model is based on three beads: C1', C2, and P. What are the parameters of the model in eq. 1? While some of the scaling parameters may be adapted from proteins, there is no P in proteins. Also, how were the equilibrium distances/angles (e.g. r0, theta0, etc in eq 1) determined? Since C1'/C2/P are not covalently bonded there is no simple way to determine those parameters e.g. from known RNA structures. Additionally, the improvements with respect to simpler models with much fewer parameters is only marginal, it is thus not clear why one should use ENCoM rather than an ANM.

2. I like the idea of using the entropic signature as features in a LASSO model to predict miRNA maturation, but the authors should explain and motivate their choice more clearly. If I understand correctly, the underlying idea is that each miRNA sequence display a given internal dynamics which is (assumed to be) tightly linked to its maturation efficiency. Since the internal dynamics could be captured by the 'entropic signature', one should be able to predict maturation efficiency from ENCoM.

While I understand and appreciate that the authors constructed a non-trivial test dataset, the resulting R2 is very low, suggesting that the model does not generalize to a sufficient degree. As such, I am not convinced that the ENCoM model can help the prediction of maturation efficiency. In order to improve the model, the authors can consider to introduce additional features to the model (e.g. the ensemble diversity as predicted from sequence, others?)

Minor points:

- the NMR bundle does not directly represent dynamics, but rather a collection of structural models that all agree to a certain extent to the measured data.

As such, NMR bundles do not necessarily represent the internal structural dynamics.

- Check several typos throughout the manuscript (chages -> changes on page 3, line 80. Diplacement -> displacement line 189 page 6. etropic -> entropic page 13 line 450, constructed -> constructed page 9 , line 286)

Reviewer #3: In this paper, the authors used a variation of the Elastic Network Model that includes sequence/context based information using atomic contacts - the ENCoM model to study the dynamics of RNA applying it to a particular class of RNA – microRNA 125a. The authors have already shown that the model works well for proteins. The paper is well structured, the analysis is thorough and covers all important aspects of model building (dataset acquisition, benchmarking, comparison to existing methods, testing, computational cost). The performance of the model given the level of coarse-graining and computational costs, is remarkable, for the pri-Mir 125a maturation efficiency predictions. However, there are a few model/parameter choices that should be explained better and a few things that should be resolved.

Eq 2 should be explained better. What do Ni and Nj represent? What do the authors mean by interaction between atoms of types? – interaction energy? What are the values for this interaction energy for the different types?

Nucleic acid folding and dynamics is inherently driven by different forces compared to proteins. The authors should provide more justification on why they chose to use the same parameter set for their ENCoM model that worked for proteins, on RNA as well? The protein parameters performs better than cut-ANM and PD-ANM, but wouldn’t RNA optimized parameter set perform even better.

Can they demonstrate that the protein parameters when applied to RNA are for example sequence sensitive? Are G-C contacts stronger than A-U contacts. Does the model with protein parameters distinguish base-pairs, wobble pairs from mismatches?

In supplementary table 1, the authors should add structures showing atom names used to assign types. e.g. Which atom in Cytosine is referred to as N5? What is the type for N1 in Uridine?

Also, how do the atom types translate to the interaction strengths in the 3 bead RNA models? It will be nice to have an illustrative example – a figure - for RNA like the authors provide in [11] for proteins.

Several RNA structures contain modified nucleotides which are structurally and functionally important. Can the authors also add stats on the numbers of structures containing modified nucleotides and the number of these altered structures that were included in their dataset and comment on the potential impact this may have on their results?

“Only the pairwise comparison between Cut-ANM and PD-ANM 363is statistically significant at p = 0:0038 despite ENCoM having the worst average 364performance of the three ENMs. The reason for this is that the two flavors of ANM 365have a similar performance profile across the 34 sequence clusters, while ENCoM tends 366to perform well where they don't and vice versa.”

Can the authors comment on what causes this difference in performance? Are there any commonalities -structural or otherwise- between the cases that ENCoM yields better results?

RMSD of 2 Angstrom seems like a very low cut off to imply that the difference is a result of a conformation change. What is the RMSD distribution for the 240 “distinct” pairs?

A few minor edits:

The colors are hard to see in Figs 7B and 7F.

Typos in page 3, line 74, 80.

**Have the authors made all data and (if applicable) computational code underlying the findings in their manuscript fully available?**

Reviewer #1: Yes

Reviewer #2: **No: **Are the parameters/examples provided online at https://nrgten.readthedocs.io/en/latest/index.html?

Reviewer #3: **No: **All data set has been provided. It will be great if the authors could provide the python code that implements the ENCoM model for RNA.

PLOS authors have the option to publish the peer review history of their article (what does this mean?). If published, this will include your full peer review and any attached files.

Reviewer #1: No

Reviewer #2: **Yes: **Sandro Bottaro

Reviewer #3: No
---

## [Decision Letter · Decision Letter 1]

19 Nov 2022

Dear Dr. Najmanovich,

Thank you very much for submitting your manuscript "Sequence-sensitive elastic network captures dynamical features necessary for miR-125a maturation" for consideration at PLOS Computational Biology. As with all papers reviewed by the journal, your manuscript was reviewed by members of the editorial board and by several independent reviewers. The reviewers appreciated the attention to an important topic. Based on the reviews, we are likely to accept this manuscript for publication, providing that you modify the manuscript according to the review recommendations.

Sincerely,

Arne Elofsson

Section Editor

PLOS Computational Biology

Arne Elofsson

Section Editor

PLOS Computational Biology

Reviewer's Responses to Questions

**Comments to the Authors:**

Reviewer #1: All previous comments have been properly addressed.

Reviewer #2: This reviewer thanks the authors for the extensive work and for providing detailed answers to all the comments.

- Would it be possible to include the link to the code in the manuscript? (I guess it is already available at https://nrgten.readthedocs.io/)

- It would be useful for potential users to include in the documentation an example relative to the application on miRNA.

- Page 6, line 199: is purine-purine stacking less stable compared to py-py stacking?

Reviewer #3: The authors have satisfactorily addressed all my comments and concerns.

**Have the authors made all data and (if applicable) computational code underlying the findings in their manuscript fully available?**

Reviewer #1: Yes

Reviewer #2: **No: **See comments to the authors

Reviewer #3: Yes

PLOS authors have the option to publish the peer review history of their article (what does this mean?). If published, this will include your full peer review and any attached files.

Reviewer #1: No

Reviewer #2: No

Reviewer #3: No

Figure Files:

Data Requirements:

Reproducibility:

References:

---

## [Editor Report · Decision Letter 2]

29 Nov 2022

Dear Dr. Najmanovich,

We are pleased to inform you that your manuscript 'Sequence-sensitive elastic network captures dynamical features necessary for miR-125a maturation' has been provisionally accepted for publication in PLOS Computational Biology.

Best regards,

Shi-Jie Chen

Academic Editor

PLOS Computational Biology

Arne Elofsson

Section Editor

PLOS Computational Biology

---

## [Editor Report · Acceptance letter]

9 Dec 2022

PCOMPBIOL-D-22-00886R2 

Sequence-sensitive elastic network captures dynamical features necessary for miR-125a maturation

Dear Dr Najmanovich,

I am pleased to inform you that your manuscript has been formally accepted for publication in PLOS Computational Biology. Your manuscript is now with our production department and you will be notified of the publication date in due course.

With kind regards,

Anita Estes
